# Solvaformer: Minimizing Geometric Redundancy for Scalable Solubility Prediction

**Jonathan Broadbent**
Sanofi, Digital R&D
Toronto, ON M5V 0E9

**Michael Bailey**
Sanofi, Digital R&D
Toronto, ON M5V 0E9

**Mingxuan Li**
Sanofi, Digital R&D
Cambridge, MA 02141

**Abhishek Paul**
Sanofi, CMC Synthetics
Cambridge, MA 02141

**Louis De Lescure**
Sanofi, CMC Synthetics
Cambridge, MA 02141

**Paul Chauvin**
Sanofi, Digital R&D
Barcelona, 08016, Spain

**Lorenzo Kogler-Anele**
Sanofi, Digital R&D
Toronto, ON M5V 0E9

**Yasser Jangjou**
Sanofi, CMC Synthetics
Cambridge, MA 02141

**Sven Jager**
Sanofi, Digital R&D
Frankfurt, 65929, Germany
sven.jager@sanofi.com

## Abstract

Accurate prediction of small molecule solubility requires balancing physical fidelity with computational scalability. While geometric deep learning offers superior inductive biases, applying full SE(3)-equivariance to dynamic multi-component systems introduces geometric redundancy and high computational cost. We introduce *Solvaformer*, a graph transformer designed to achieve simplicity at scale by selectively grounding interactions in geometry. The architecture challenges the need for global equivariance: it applies strict SE(3)-equivariant attention only to rigid *intramolecular* structures, while modeling fluid *intermolecular* interactions via computationally efficient scalar attention. While Solvaformer demonstrates strong performance (approaching the DFT baseline), we also report that a simpler MPNN augmented with physics-informed partial charges (MLIPs) can achieve similar performance. This suggests that for scalar solubility prediction, high-quality electronic descriptors may render end-to-end equivariant architectures redundant. Our findings highlight that geometric redundancy can be minimized either architecturally (Solvaformer) or via decoupled feature generation (MPNN w/ MLIPs), offering two scalable paths for solution-phase modeling.

## 1 Introduction

Efficient pharmaceutical manufacturing relies on optimizing synthesis pathways, a process heavily dependent on finding the right solvents to dissolve transient intermediate molecules. Because these intermediates are novel and scarce, experimental solubility testing is limited, creating a critical need for accurate predictive models. Standard models struggle to generalize to these unseen chemical spaces, however to our knowledge there is not yet a geometric model for solubility (de Ruyter et al., 2022; Byrne et al., 2016; Zhang et al., 2021). This challenge presents an opportunity to address the workshop's theme of minimizing geometric redundancy: determining whether modeling solution-phase chemistry requires computationally expensive 3D-equivariant architectures or if simpler methods suffice.

Existing approaches to solubility prediction span a spectrum from physics-based to purely data-driven. DFT-based models (e.g., COSMO-RS (Klamt, 2005)) provide highly accurate thermodynamic predictions but require up to 10 hours per molecule for conformer generation and optimization, rendering them impractical for high-throughput screening (Kastenholz & Hünenberger, 2006). In contrast, Message Passing Neural Networks (MPNNs) (Gilmer et al., 2017) achieve competitive performance by operating on 2D molecular graphs, offering scalability but lacking explicit 3D reasoning and interpretability. To bridge these gaps, we build upon Equiformer (Thomas & Smidt,

2022), an SE(3)-equivariant graph transformer that respects rotational and translational symmetries through spherical harmonic representations (see Appendix E.3 for a mathematical description). This architecture provides a foundation for our Solvaformer design, which we adapt to handle multi-component solution-phase system.

We investigate two distinct strategies to minimize geometric redundancy. First, we introduce Solvaformer, a hybrid transformer that applies strict SE(3)-equivariance only to rigid intramolecular structures while relaxing fluid solute-solvent interactions to efficient scalar attention. Second, we challenge the need for architectural geometry entirely by evaluating a simpler MPNN augmented with physics-informed charges (machine learned interatomic potentials; MLIPs). Our work contributes a rigorous comparison of these "architectural" versus "feature-based" approaches, demonstrating that eliminating redundant geometric constraints improves both scalability and performance in large-scale solubility prediction.

## 2 METHODS

### 2.1 DATA

To balance experimental relevance with computational scale, we trained our models on a combined dataset of experimental solubility measurements and quantum-mechanical calculations. We utilized BigSolDB 2.0 (Krasnov et al., 2025), comprising 103,944 experimental LogS values across 213 solvents (243–425 K), and CombiSolv-QM (Vermeire & Green, 2020), providing 1 million COSMO-RS solvation free energies ($\Delta G_{\text{solv}}$). Solvaformer was trained using a multi-task alternating-batch strategy to predict both experimental LogS and theoretical $\Delta G_{\text{solv}}$, enabling generalization across broad chemical space while leveraging the consistency of large-scale QM data.

### 2.2 SOLVAFORMER

In our work, we modify the EquiformerV2 architecture to enable solubility prediction. Instead of receiving one molecule as input, Solvaformer receives multiple molecules (a solute and one or more solvents). EquiformerV2 only uses equivariant attention, whereas in Solvaformer there are two types of attention: equivariant attention between intramolecular atoms and scalar attention between intermolecular atoms.

Challenging the assumption that full SE(3)-equivariance is required for all atomic pairs, we recognize that the relative spatial relationship between solute and solvent is stochastic and effectively undefined. Modeling this with equivariant kernels would introduce unnecessary complexity. Instead, we enforce independent SE(3) symmetries for each molecule. This simplifies the learning problem: we use computationally intensive equivariant attention only where geometry is rigid (intramolecular) and efficient scalar attention (i.e. keys and queries are computed using *only* the scalar part of node features) where geometry is fluid (intermolecular). This design embodies the principle of simplicity at scale, directing compute budget only to where geometric grounding provides inductive bias.

Equivariant and scalar attention modules aggregate messages from within-molecule and from other-molecules. Suppose $i$ indexes a destination atom, $j$ indexes another atom in the same molecule, and $\zeta$ indexes an atom in the other molecule, with embeddings $x_i$, $x_j$, and $x_\zeta$. To compute the message incident on atom $i$, we compute the messages from equivariant attention and scalar cross-attention, and then sum them. For the equivariant attention, Equiformer computes message tensor values $v_{ij}^e$ using tensor products between $x_i$ and $x_j$, and similarly projects from $x_i \otimes x_j$ down to scalar features $f_{ij}^{(0)}$, which it then passes through a layer norm and activation to produce a logit

$$z_{ij}^e = \text{LeakyRELU}(\text{LayerNorm}(f_{ij}^{(0)})). \tag{1}$$

The scalar cross-attention is a simple implementation of traditional dot product attention, with the message values $v_\zeta^s$, and the key and query vectors, $k_\zeta$ and $q_i$ computed from the scalar part of the embedding, $x_\zeta^{(0)}$, by linear maps. Then the logits are $z_{i\zeta}^s = \langle q_i, k_\zeta \rangle$. The messages are aggregated the same way for both, with the caveat that equivariant messages are tensorial, while scalar messages

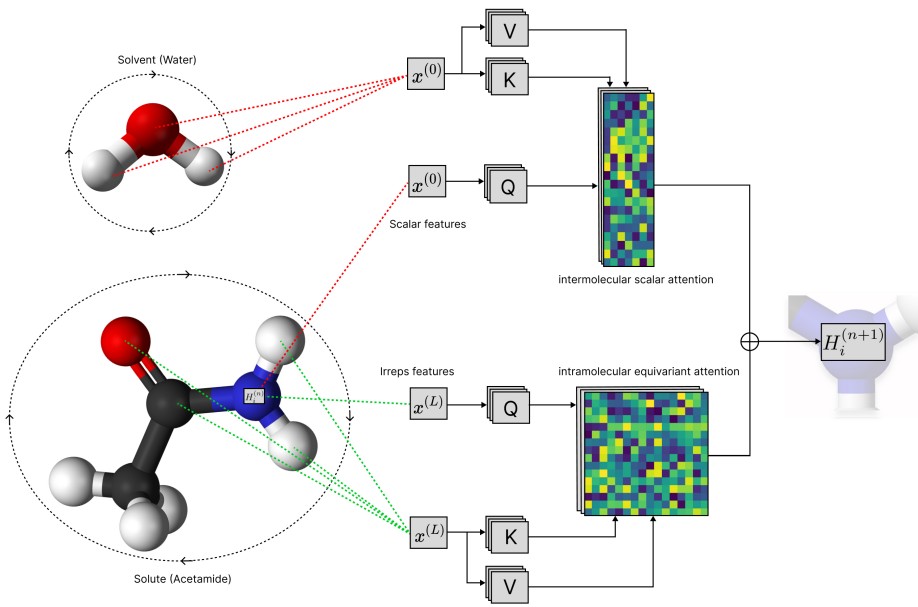

Figure 1: Example of Solvaformer performing an update of the hidden representation of a nitrogen atom ($H_i$) in a single layer for inputs water and acetamide.

are purely scalar quantities:

$$m_i^e = \sum_j \text{softmax}_j(z_{ij}^e) \ v_{ij}^e \qquad m_i^s = \sum_\zeta \text{softmax}_j(z_{i\zeta}^s) \ v_{i\zeta}^s \tag{2}$$

$$\mu_i = m_i^e + m_i^s \tag{3}$$

### 2.3  MPNN AUGMENTED WITH MLIPs

We adopted a standard Gated Graph Neural Network (GG-NN) architecture adapted for solubility prediction (Gilmer et al., 2017), which aggregates solute and solvent features via an interaction map before concatenating them with temperature for final prediction. To test if geometric information can replace geometric architecture, we augmented this scalar baseline with physics-informed descriptors. We utilized the NVIDIA ALCHEMI microservice with the AIMNet2 backbone (Anstine et al., 2025) to predict partial atomic charges from 3D structures. Because AIMNet2 compresses geometric features (distances and $l = 1$ harmonics) into invariant scalars, this approach injects high-fidelity electronic information into the MPNN without incurring the computational cost of explicit equivariant message passing or traditional DFT.

## 3  RESULTS

Table 1 summarizes the predictive performance of the evaluated models on the BigSolDB 2.0 test set. For context, we include *XGBoost-DFT* as a physics-informed upper bound; it achieves the lowest error (MAE 0.621, RMSE 0.806) but relies on computationally expensive DFT calculations that render it impractical for high-throughput screening (see Appendix A for runtime analysis). The remaining learned models are analyzed below, categorized by their incorporation of geometric features and equivariant processing.

**Non-Geometric Baselines.** The models lacking both geometric features and equivariance generally exhibited the highest errors. *SolvBERT* performed poorest (MAE 0.871, RMSE 1.110), suggesting that end-to-end language modeling on SMILES strings struggles to capture solubility determinants without explicit structural biases. Fingerprint-based methods (*XGBoost-CFP*) improved upon this (MAE 0.749), and leveraging large-scale pre-trained embeddings in *XGBoost-MMB* further reduced

Table 1: Model performance metrics on BigSolDB 2.0 test set

| Model | Geometric features | Equi-variance | MAE | MSE | RMSE | $R^2$ | Pearson | Spearman |
|---|---|---|---|---|---|---|---|---|
| *XGBoost-DFT* | ✓ | ✗ | 0.621 | 0.650 | 0.806 | 0.499 | 0.722 | 0.722 |
| SolvBERT | ✗ | ✗ | 0.871 | 1.231 | 1.110 | 0.052 | 0.247 | 0.222 |
| XGBoost-CFP | ✗ | ✗ | 0.749 | 0.889 | 0.943 | 0.315 | 0.592 | 0.561 |
| Solvaformer w/ eqIMA | ✓ | ✓ | 0.741 | 0.900 | 0.949 | 0.306 | 0.672 | 0.640 |
| XGBoost-MMB | ✗ | ✗ | 0.710 | 0.831 | 0.912 | 0.360 | 0.616 | 0.591 |
| Solvaformer w/o IMA | ✓ | ✓ | 0.691 | 0.858 | 0.926 | 0.345 | 0.626 | 0.606 |
| MPNN | ✗ | ✗ | 0.668 | 0.746 | 0.864 | 0.425 | 0.724 | 0.693 |
| Solvaformer | ✓ | partial | 0.643 | 0.700 | 0.837 | 0.460 | 0.694 | 0.677 |
| MPNN w/ MLIPs | ✓ | ✗ | 0.629 | 0.667 | 0.817 | 0.486 | 0.721 | 0.710 |

IMA: intermolecular attention; MLIP: machine learning interatomic potentials

error to an MAE of 0.710. However, the standard *MPNN*, which operates on the 2D molecular graph without 3D features, outperformed all other non-geometric approaches, achieving an MAE of 0.668 and an RMSE of 0.864.

**Geometric and Equivariant Models.** Introducing 3D geometry led to further performance gains. We first examined *Solvaformer w/o IMA*, an ablation where the scalar intermolecular attention is removed. Despite possessing full SE(3)-equivariance and geometric features, this variation underperformed the simple MPNN (MAE 0.691, RMSE 0.926), indicating that equivariant processing without intermolecular communication is insufficient. However, the full **Solvaformer** architecture—which combines geometric features with *partial* equivariance (intramolecular SE(3)-attention + intermolecular scalar attention)—substantially improved performance, lowering the MAE to 0.643 and RMSE to 0.837. This confirms the value of the hybrid architectural design. Finally, the **MPNN w/ MLIPs** model, which incorporates high-fidelity geometric information via AIMNet2-derived partial charges but does *not* use an equivariant architecture, achieved the lowest error among all learned methods (MAE 0.629, RMSE 0.817). This model effectively matches the performance of the computationally intensive XGBoost-DFT baseline while avoiding the complexity of explicit equivariant message passing.

## 4 DISCUSSION

**Is Geometry Redundant?** Our results offer a nuanced answer. The superior performance of the MPNN w/ MLIPs baseline challenges the assumption that equivariant architectures are strictly required for scalar property prediction. While Solvaformer minimizes redundancy architecturally via scalar intermolecular attention, the MPNN pipeline suggests we can decouple the geometric engine entirely. Because AIMNet2 is geometric but invariant (predicting scalar charges without propagating equivariant tensors) the MPNN pipeline operates without any SE(3)-equivariant layers. This supports a "Simplicity at Scale" paradigm where expensive geometric inference is performed only once for feature generation.

However, simplicity is multifaceted. While the MPNN pipeline is computationally efficient, Solvaformer offers operational simplicity as a single end-to-end model, avoiding multi-stage inference workflows. Furthermore, Solvaformer enables interpretability that scalar models lack; its intermolecular attention maps successfully isolate physical phenomena like hydrogen bonding (Appendix B). Thus, the choice represents a trade-off between computational minimality (MPNN) and mechanistic interpretability (Solvaformer).

**Future Work.** Our investigation may be limited by fidelity of data preprocessing. Both approaches presented here rely on static 3D conformers generated via MMFF minimization, which optimizes for gas-phase stability rather than condensed-phase solubility. This mismatch may obscure the benefits of stricter geometric methods, as equivariant architectures might only show clear dominance when provided with accurate solution-phase structures. Furthermore, molecules in solution exist as dynamic

Boltzmann ensembles rather than single rigid conformers (Cordova et al., 2024). Since computing accurate solution-phase ensembles at scale remains computationally prohibitive, developing efficient ensemble modeling methods is a critical frontier for elucidating *in-silico* solution phase modeling. This is briefly explored in Appendix C.

**Takeaway.** This study addresses the workshop's theme of scale and simplicity by demonstrating that effective geometric deep learning requires carefully distinguishing between essential and redundant structure. By retaining strict SE(3)-equivariance for intramolecular features while adopting a simplified scalar attention for intermolecular interactions, Solvaformer respects the physical reality that solution-phase relative geometry is stochastic. Our ablation studies support that this hybrid design is effective. Moreover, our results with the MLIP-augmented MPNN demonstrate that geometric redundancy can be minimized even further by offloading 3D reasoning to pre-trained potentials, validating that simpler scalar architectures can suffice when supported by physics-informed descriptors. Ultimately, both approaches offer distinct advantages (computational efficiency versus mechanistic interpretability) validating that targeted geometric grounding is a practical path for high-throughput material modeling.

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

## A   RUNTIME ANALYSIS

Runtime analysis reveals critical scalability differences among models for small molecule solubility prediction. MPNN and Solvaformer maintain low inference latencies suitable for synthesis optimization, while DFT-based methods prove prohibitive.

**Methods.**   We conducted a runtime analysis of MPNN, Solvaformer, XGBoost-CFP, and XGBoost-DFT on the BigSolDB 2.0 dataset, measuring end-to-end inference time (SMILES to LogS output, including conformer generation) as a function of batch size and molecular size. For each data point, we sampled five repeats from BigSolDB 2.0, using replacement when constraints (e.g., molecules with 180–220 atoms) yielded insufficient samples. Measurements focused on inference to simulate real-world use cases where users require rapid predictions for pathway optimization; training costs are one-off and thus less constraining.

Solvaformer used a batch size of 32 and MPNN a batch size of 3000 to maximize GPU utilization without memory overflows on large molecules. To avoid excessive DFT computations, we profiled five molecules spanning molecular weights and fit an empirical formula to estimate XGBoost-DFT preprocessing time based on solute-solvent molecular weight.

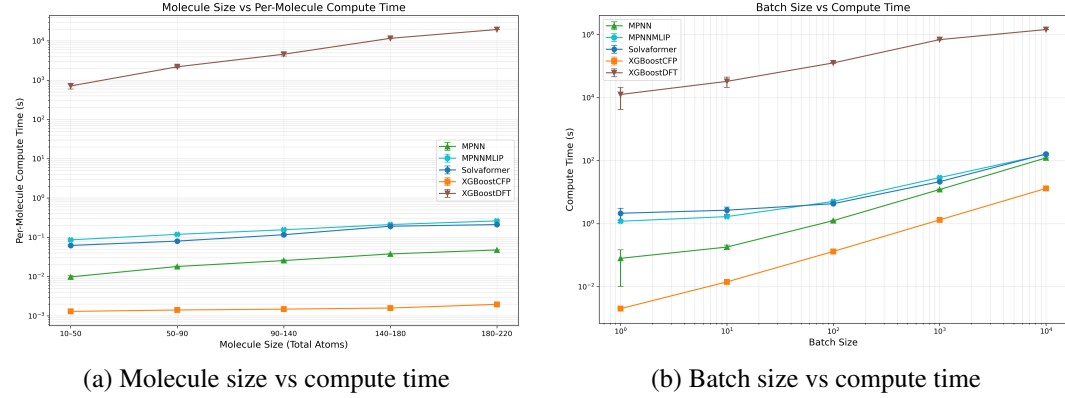

(a) Molecule size vs compute time      (b) Batch size vs compute time

Figure 2: Runtime analysis of compute time for small-molecule solubility prediction methods. Each method was computed on 4 CPUs and 1 Nvidia H100 GPU (80GB)

**Key Findings.** XGBoost-CFP achieves near-instant predictions ($< 10^{-2}$ s per sample), highlighting the efficiency of 2D fingerprints. In stark contrast, XGBoost-DFT requires $10^3$-$10^4$ s per prediction ($\sim$30 minutes to 3 hours), dominated by DFT preprocessing and rendering it inaccessible despite superior fidelity.

Runtime scaling with molecular size further discriminates methods: XGBoost-DFT exhibits exponential growth due to quantum chemistry demands, while MPNN, Solvaformer, and XGBoost-CFP increase only marginally. GPU-accelerated models (MPNN, Solvaformer) leverage batching effectively—runtimes remain flat from batch size 1 to 10 (near-optimal GPU occupancy), then scale linearly beyond the model's native capacity.

## B  EXPLAINABILITY CASE STUDY: DISTINGUISHING INTRA- VS. INTERMOLECULAR HYDROGEN BONDS

An advantage of Solvaformer is its ability to interpret the complex 3D relationships that govern molecular interactions. To provide a compelling demonstration, we analyzed the model's inter molecular attention maps for the solubility of two isomers in water: salicylic acid (*ortho*-hydroxybenzoic acid) and 4-hydroxybenzoic acid (*para*-hydroxybenzoic acid). This pair represents a classic chemical challenge where the change in substituent position dictates whether hydrogen bonding is internal (intramolecular) or external (intermolecular).

The key atom for this comparison is the hydroxyl proton, labeled H15 in our structures. In 4-hydroxybenzoic acid, H15 is exposed and free to form intermolecular hydrogen bonds with water, enhancing solubility. In salicylic acid, however, the adjacent geometry allows H15 to form a strong intramolecular hydrogen bond with the carbonyl oxygen. This internal bond makes H15 unavailable for solvent interactions, thus lowering solubility. The attention maps in Figure 3 show that Solvaformer captures this distinction.

For the *para* isomer (Figure 3b), the hydroxyl proton H15 is geometrically unhindered. The attention map correctly reflects this by showing a distinct interaction between H15 and the atoms of the water molecule. This signal is direct evidence that the model identifies H15 as an active site for intermolecular hydrogen bonding.

In stark contrast, the attention map for salicylic acid (*ortho*) (Figure 3d) shows that the attention between the hydroxyl proton H15 and the solvent is effectively zero. This absence of interaction is the critical finding. It demonstrates that the model has learned that H15 is "occupied" by the intramolecular hydrogen bond with the nearby carbonyl oxygen and is therefore unavailable to bond with water.

This case study proves that Solvaformer is not merely correlating features but is learning physically meaningful, 3D-aware principles of solvation chemistry. The ability to distinguish between competing

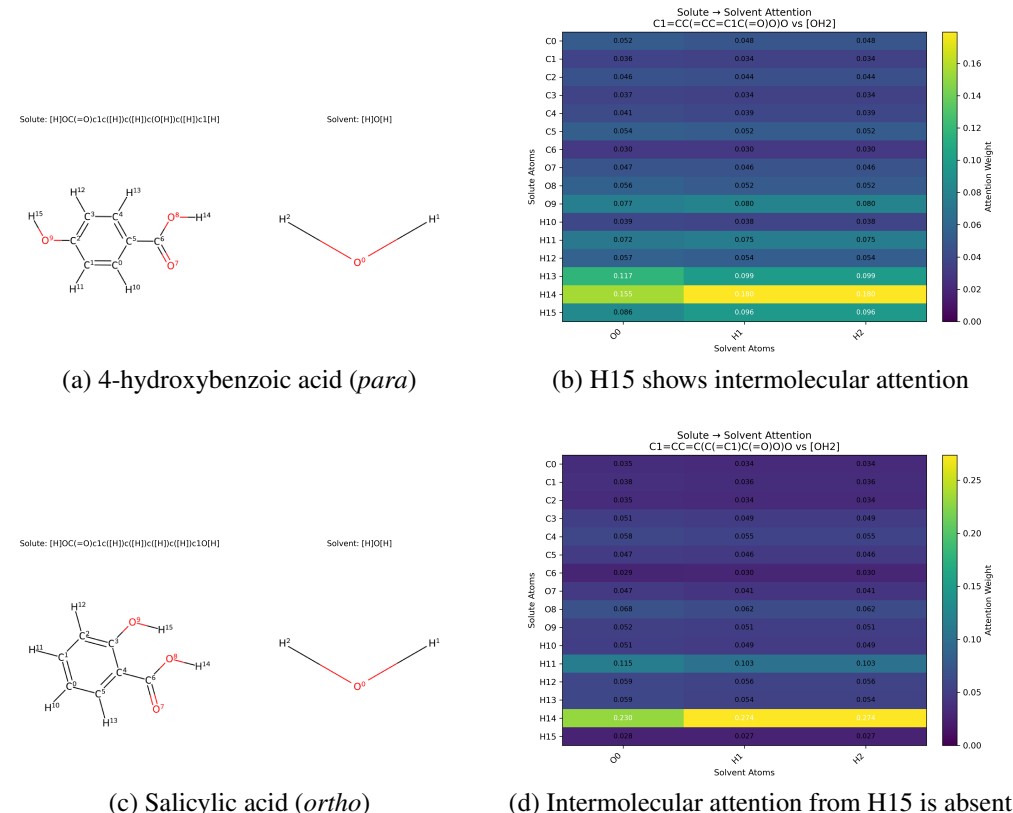

(a) 4-hydroxybenzoic acid (*para*)

(b) H15 shows intermolecular attention

(c) Salicylic acid (*ortho*)

(d) Intermolecular attention from H15 is absent

Figure 3: Solute-to-solvent attention maps demonstrating Solvaformer's chemical intuition. (b) For the *para* isomer, the hydroxyl proton (H15) shows clear attention to water, indicating intermolecular H-bonding. (d) For the *ortho* isomer, this attention from H15 disappears, correctly implying it is occupied in a dominant intramolecular H-bond.

bonding scenarios based on geometry is a sophisticated piece of chemical reasoning that makes the model's predictions both more accurate and highly interpretable

## C  Solution Phase Conformers

One drawback in our current model is that the 3D conformers we generate are not accurate to their respective solvent environment. As discussed in Section 4, molecules in solution exist as dynamic Boltzmann ensembles rather than single static conformers, and gas-phase MMFF minimization may obscure the inductive biases that equivariant architectures are designed to exploit. To investigate whether solution-phase conformers can improve Solvaformer performance, we evaluate three conformer generation strategies that vary in their degree of solvent awareness.

**Methods.**  The first method, *GNNImplicitSolvent* (GNNIS) (Katzberger et al., 2025), combines OpenMM molecular simulation with a graph neural network to optimize 3D conformers into low-

| Conformer Method | solvent dependent | MAE | MSE | Pearson | Spearman |
|---|---|---|---|---|---|
| GNNImplicitSolvent | ✓ | 0.6482 | 0.7143 | 0.7651 | 0.7462 |
| MMFF-Ensemble | ✓ | 0.8701 | 1.3058 | 0.6211 | 0.5863 |
| MMFF | ✗ | 0.7905 | 1.0527 | 0.6789 | 0.6682 |

Table 2: Solvaformer performance on BigSolDB 2.0 subset using alternative conformer generation methods

energy configurations within a specified solvent environment. Multiple conformers are first generated using RDKit `EmbedMultipleConfs`, then minimized by the GNNIS model, filtered to retain those within the lowest energy decile (up to ten), and further diversity-filtered via Butina clustering on conformer RMSD ($\delta = 0.15$ Å). Because GNNIS is limited to 39 solvents, we subset BigSolDB 2.0 accordingly for all three comparisons.

The second method, *MMFF-Ensemble*, follows the same ensemble pipeline but replaces the GNNIS minimizer with RDKit MMFF94s augmented with the experimental dielectric constant of the target solvent. This provides an intermediate level of environmental awareness at a fraction of the computational cost of GNNIS. The third method, *MMFF*, serves as the control and mirrors the default Solvaformer preprocessing: a single conformer generated and minimized by MMFF94s without any solvent-specific parameterisation.

For all ensemble methods, each `PyTorch Data` object stores a tensor of conformer positions alongside their associated energies. During training, a single conformer is drawn per sample by sampling from the Boltzmann distribution,

$$p_i \propto \exp\left(-\frac{e_i}{k_B T}\right),$$

where $e_i$ is the relative energy of conformer $i$, $T$ is the experimental measurement temperature, and $k_B$ is the Boltzmann constant. To account for stochastic inference, predictions at validation and test time are aggregated over $n = 10$ and $n = 50$ forward passes per sample, respectively.

**Results.** GNNIS achieves the strongest performance (MAE $0.648 \pm 0.04$), improving over the MMFF control (MAE $0.791$) by approximately 18% and yielding the rank correlations (Pearson $0.765$, Spearman $0.746$). This confirms that grounding conformers in solution-phase energetics provides a meaningful signal for Solvaformer. The MMFF-Ensemble method, however, underperforms both alternatives (MAE $0.870$), failing to realise the expected benefit of ensemble diversity. We attribute this in part to early stopping: the model was trained with patience 20 and $\Delta_{\min} = 0.01$, and MMFF-Ensemble converged before the model had sufficiently learned the conformer distribution—an effect that was less pronounced for GNNIS, whose higher-quality energies provide a sharper Boltzmann signal from the outset.

**Limitations.** Despite its accuracy advantage, GNNIS is impractical for large-scale deployment. Generating solution-phase ensemble conformers via OpenMM simulation introduces per-molecule latency that is orders of magnitude higher than MMFF, making real-time or high-throughput inference infeasible. Developing lightweight surrogate methods—such as learned implicit-solvent force fields or fast diffusion-based conformer samplers—that can approximate solution-phase ensembles at MMFF speeds is therefore an important direction for future work.

## D    DATA PREPROCESSING

### D.1    BIGSOLDB 2.0

We utilized the BigSolDB 2.0 dataset, a comprehensive solubility resource comprising 103,944 experimentally measured solubility values for 1,448 unique organic solutes in 213 solvents, across a temperature range of 243–425 K. These values were manually curated from 1,595 peer-reviewed publications and standardized into a machine-readable format including SMILES representations for both solutes and solvents. LogS values (log molar solubility in mol/L) were calculated using solvent densities either from experimental measurements or interpolated via linear models where necessary. The dataset spans aqueous and non-aqueous solvents, including common organic media such as ethanol, acetone, and ethyl acetate, enabling broad coverage for solubility prediction tasks (Krasnov et al., 2023; 2025).

To ensure the quality of the data used for model development, we applied the following filtering criteria:

- Canonicalized the SMILES of both solutes and solvents using RDKit.

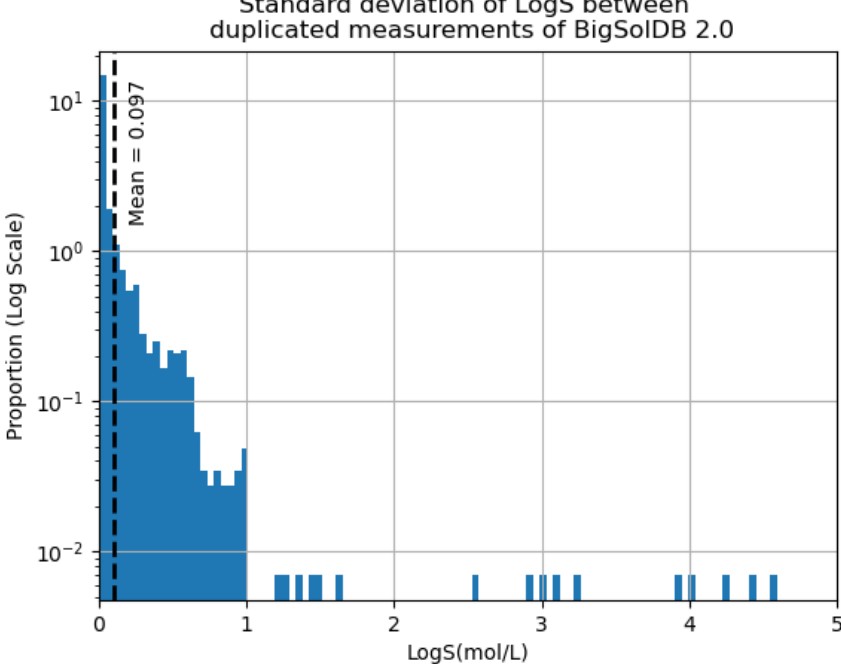

Figure 4: 6591 measurements in BigSolDB 2.0 had duplicated measurements from separate sources (same solute, solvent, temperature but measured in a different laboratory and have different measured solubility). We removed these measurements from the dataset. Here we measure the standard deviation within groups of duplicated measurements and plot the distribution. This provides an estimate of the precision of experimental measurements for solubility and hence lower bound for error rate prediction of our dataset.

- Removed entries containing bimolecular solutes or multi-component species.
- Excluded all metal-containing and ionic solute entries.
- Discarded entries lacking a LogS value.
- Discarded all duplicate entries.

The dataset had 6591 duplicated entries. LogS between duplicated entries had an average standard deviation of $0.0974$, which represents an intrinsic limit to the performance of our models (Figure 4). Following data filtering, we split the dataset into training and test sets using chemical space-aware clustering:

- Solutes were clustered using the Butina algorithm based on Tanimoto similarity of Morgan fingerprints (radius = 2).
- From these clusters, we sampled solutes across the chemical space to form a structurally diverse test set (10% of data).

The final split consisted of 82,758 solute-solvent measurements for training and 9,250 for testing, with 1,142 unique solutes in the training set and 126 in the test set (See figures 6, 5).

This stratified, diversity-aware split enables robust benchmarking of model generalization to solute structures.

### D.2 COMBISOLV

The CombiSolv-QM dataset (Vermeire & Green, 2020) provides quantum-mechanically computed solvation free energies for approximately one million solvent–solute pairs. These values were derived

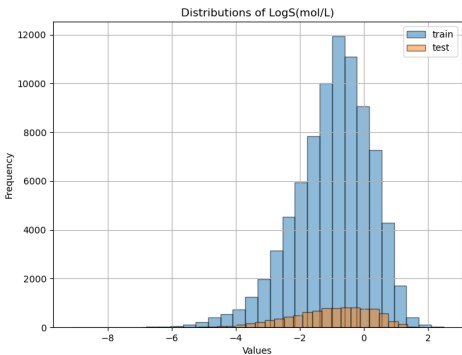

Figure 5: Distribution of measured logS in the train-test split.

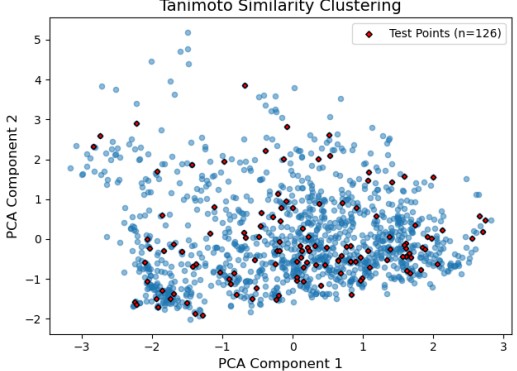

Figure 6: Butina clustering of tanimoto similarity of all unique solutes in BigSolDB 2.0

using COSMO-RS theory via the COSMOtherm software, covering 11,029 solutes and 284 solvents. All solvation energies ($\Delta G_{\mathrm{solv}}$) were calculated at 298 K using a conformer-aware protocol that includes DFT-based geometry optimization followed by chemical potential analysis in solution.

In our work, CombiSolv-QM was used in conjunction with the BigSolDB 2.0 dataset to train the Solvaformer model. To enable learning from both experimental and computational data, we implemented an alternating batch training scheme: each mini-batch was sampled from either CombiSolv-QM or BigSolDB 2.0. The model was trained with two separate prediction tasks: one for experimental solubility values (LogS) and one for calculated solvation free energies ($\Delta G_{\mathrm{solv}}$). The training set of CombiSolv-QM was filtered for solutes in the BigSolDB 2.0 test set.

This dual-target, alternating-batch strategy allowed the model to generalize across both experimental and theoretical chemical space while maintaining fidelity to each target type. It also served as an effective form of multi-task learning, allowing the model to benefit from the scale and consistency of CombiSolv-QM and the real-world relevance of BigSolDB 2.0.

# E  ADDITIONAL MODEL DETAILS

## E.1  SOLVBERT

SolvBERT is a transformer-based model that treats solute–solvent complexes as combined SMILES sequences, applying NLP-style encoding to molecular interactions (Yu et al., 2023). Unlike graph-based models that process solute and solvent separately, SolvBERT ingests the concatenated SMILES of the complex and converts them into contextualized embeddings using a BERT backbone pretrained in an unsupervised masked language modeling manner on a large computational dataset (CombiSolv-QM) (Yu et al., 2023). With this setup, the self-attention network is able to learn interactions between solute and solvent. Following pretraining, the model is fine-tuned either on experimental solvation free energy or solubility datasets, demonstrating strong performance across both tasks.

Empirical evaluations show that SolvBERT achieves solvation free energy predictive accuracy comparable to state-of-the-art graph-based models like MPNN. It also surpasses hybrid graph-transformer architectures such as GROVER when predicting solubility on out-of-sample solvent–solute combinations (Yu et al., 2023). The unsupervised pretraining enables better internal clustering of molecular systems (via TMAP visualization), supporting enhanced generalization despite varied fine-tuning targets.

## E.2  XGBOOST MODELS

We use XGBoost to predict solubility using a variety of embedding methods. For **DFT** features, we first generated conformers using RDKit ETKDGv3 through WEASEL 1.12. Conformers were then optimized using GFN2-xTB, and the five most stable conformers (or those covering 90% of the Boltzmann population at the xTB level) were subsequently subjected to DFT calculations at the wB97X-V/def2-TZVP level of theory in the gas phase. The energies derived from these DFT calculations were used to apply Boltzmann weights to the resulting molecular features. Features were calculated for both the solute and the solvent molecules. The most stable conformer was further evaluated with a COSMO-RS calculation using ORCA 6.0(Neese, 2012) to obtain the free energy of solvation.

We also generated **Circular Fingerprints (CFP; Morgan fingerprints)**(Morgan, 1965) with RDKit, using a radius of 2 and a fingerprint size of 2048 bits, with count simulation disabled, chirality excluded, bond types enabled, ring-membership information included, default count bounds, and without restricting to nonzero invariants.

In addition, we produced learned embeddings using **MegaMolBART (MMB)**, a large language model trained on 1.5 billion SMILES from the ZINC-15 dataset (Irwin et al., 2022; NVIDIA Corporation, 2024) and previously shown to be effective for molecular property prediction (Moayedpour et al., 2024). From the pretrained model we extracted 512-dimensional embeddings and used them directly as inputs to XGBoost.

Furthermore, the **SolvBERT** model was initially trained for temperature independent solubility. To adapt it for our purposes we first trained the model on the CombiSolv and BigSolDB 2.0 training data (Training followed the procedure described here: https://github.com/su-group/SolvBERT (Yu et al., 2023) and then took the embeddings from the pre-trained model and trained an XGBoost regressor with temperature as an additional feature to the model.

Using these feature sets, we trained a standard XGBoost regressor (Chen & Guestrin, 2016) with `n_estimators`=500, `learning_rate`=0.1, and `max_depth`=6.

Table 3: DFT calculated molecular features

| Feature name | Feature Description |
|---|---|
| Dipole_Moment_Debye | Dipole moment in Debye |
| LUMO_E_Eh | Energy of Lowest unoccupied molecular orbital (LUMO) in Hartrees |
| LUMOX_E_Eh | Energy of LUMO - X in Hartrees |
| HOMO_E_Eh | Energy of Lowest occupied molecular orbital (HOMO) in Hartrees |
| HOMOX_E_Eh | Energy of HOMO - X in Hartrees |
| HOMO_LUMO_gap | Energy difference between HOMO and LUMO |
| Dispersion_correction | Dispersion correction calculated with VV10 nonlocal van der Waals correlation |
| Cavity_Volume | CPCM cavity volume in cubic angstroms |
| Cavity_Surface_area | CPCM solvent-accessible surface in squared angstroms |
| Surface_Charge_CPCM | Total apparent surface charge distribution calculated by CPCM |
| C_charge_total | Sum of all Hirshfeld charge on all carbon atoms |
| O_charge_total | Sum of all Hirshfeld charge on all oxygen atoms |
| N_charge_total | Sum of all Hirshfeld charge on all nitrogen atoms |
| H_charge_total | Sum of all Hirshfeld charge on all hydrogen atoms |
| Het_charge_total | Sum of all Hirshfeld charge on all heteroatoms |
| energy_kcal_mol | Electronic energy of the system in kcal/mol |
| dGs | Free energy of solvation as calculated by open COSMO-RS through ORCA6 |

### E.3 EQUIFORMER

Equiformer is analogous to an ordinary graph transformer in the following sense:

- Instead of weights and activations taking scalar values, they take values in an SO(3) representation space. These representations are equivalent to *spherical harmonics* (also known as *orbitals*), so a weight or activation can be seen as an approximated function on the sphere $S^2$.
  - When a representation vector $f$ is decomposed into irreducible representations (i.e., different angular frequencies) $f_\ell$, its 'rotations' correspond to Wigner D-matrices:
  $$f_\ell \mapsto D^\ell(R)\, f_\ell$$
  - Multiplication of these SO(3) representations corresponds to multiplication of their spherical functions (dropping high-frequency terms where needed)
- In addition to taking non-scalar values, the weights are also spatially varying *functions*, depending on the relative vector between the communicating nodes. The spatial variation of these weights is also represented using a spherical harmonic decomposition, with radial dependence.
  - Therefore, the weight functions (and thus the model) are *equivariant* if rotating the evaluation vector in 3D space corresponds to "rotating" the weight value.

To compute the product of spherical functions $f$ and $g$ with harmonic decompositions $f_{\ell_1,m_1}$ and $g_{\ell_2,m_2}$, Equiformer uses tensor products based on Clebsch–Gordan coefficients (Sharp, 1960) $C_{\ell_1,\ell_2}^{\ell_3}$, which combine the components in the correct way:

$$[f_{\ell_1} \otimes g_{\ell_2}]_{\ell_3,m_3} = \sum_{m_1,m_2} C_{\ell_1,m_1;\,\ell_2,m_2}^{\ell_3,m_3}\, f_{\ell_1,m_1}\, g_{\ell_2,m_2}. \tag{4}$$

Of course, what distinguishes Equiformer from an ordinary equivariant message passing network is that Equiformer uses the above operations to build an equivariant attention mechanism, so that each node and head can pay different amounts of attention to different neighbor nodes.

EquiformerV2 (Liao et al., 2024) enhances the original Equiformer architecture. It replaces the SO(3)-equivariant convolutions with eSCN convolutions, reducing computational complexity from $O(L_{max}^6)$ to $O(L_{max}^3)$, enabling scaling to higher-degree ($L = 6$) representations (Passaro & Zitnick, 2023; Liao et al., 2024).

EquiformerV2 achieved state-of-the-art results on large-scale datasets (e.g., OC20/OC22), which use force and energy of individual molecules as the training target. However, EquiformerV2 is not equipped to predict solubility.

### E.4  MPNN

We adopted an MPNN architecture variation (gated graph neural network; GG-NN+set2set) specifically developed for molecular property prediction (Gilmer et al., 2017) by modifying the GG-NN message passing architecture. The model consists of three stages, namely the message-passing phase followed by an interaction phase, and finally the prediction phase. In the message-passing phase, features of the immediate neighboring nodes are aggregated into a node's contextual information via unidirectional edge networks. This process is repeated for *n* message passing steps, which is a hyperparameter tuned when training our model. In the interaction phase, an interaction map is built by performing a matrix multiplication on the aggregated solute and solvent feature tensors. Solute-solvent interactions are then resolved by mapping solute features on to the solvent tensor and vice versa. Finally, the updated solute and solvent tensors are concatenated along with 'temperature' as an external input to create the final features' tensor. The final features' tensor is passed through three ReLU activation layers before correlating with the target, logS in this case.

To assess whether explicit geometric architecture is strictly necessary or if geometric information suffices, we augmented the MPNN input with partial atomic charges. We utilized the NVIDIA ALCHEMI inference microservice to compute these charges for all solute and solvent molecules. ALCHEMI functions as a high-throughput container for machine learning interatomic potentials (MLIPs); specifically, we employed the AIMNet2 backbone, which predicts nearly quantum-accurate partial charges directly from an 3D structure. Node features are initialized with geometric information (interatomic distances and $l = 1$ harmonics) yet compresses them to scalars between layers such that message passing is invariant (Anstine et al., 2025). This approach enabled the rapid annotation of our entire dataset with electronic descriptors without incurring the prohibitive computational cost of traditional DFT-based charge assignment (e.g., RESP or Hirshfeld), effectively bridging the gap between scalar graph features and geometry-informed electronic properties.

### E.5  Solvaformer Training

We trained Solvaformer, using a combined dataset of BigSolDB 2.0 and CombiSolv-QM, sampled in equal ratios via alternating batches. 3D conformers were generated using RDKit and minimized by Merck molecular force field (MMFF). The model was trained with a batch size of 6 for up to 100 epochs, with early stopping based on a patience of 20 and a minimum delta of 0.01. Solvaformer consists of 8 layers with 8 attention heads, 128-dimensional spherical channels, and 96-dimensional hidden dimensions in both attention and feedforward networks. It uses SE(3)-equivariant operations with angular momentum up to $l = 6$, and includes solvent-solvent attention and edge features. Regularization includes alpha dropout (0.5), drop path (0.4), and projection dropout (0.4). The model predicts both solubility and solvation energy using separate outputs and is optimized with mean squared error loss and a learning rate of $3 \times 10^{-6}$. All hyperparameters were selected based on a hyperparameter optimization experiment, the details of which are provided in the appendix (Figure 7).

## F  Data Availability

All the raw data used to train and test the models is publicly available and can be found here:

- BigSolDB2.0 (Krasnov et al., 2025)

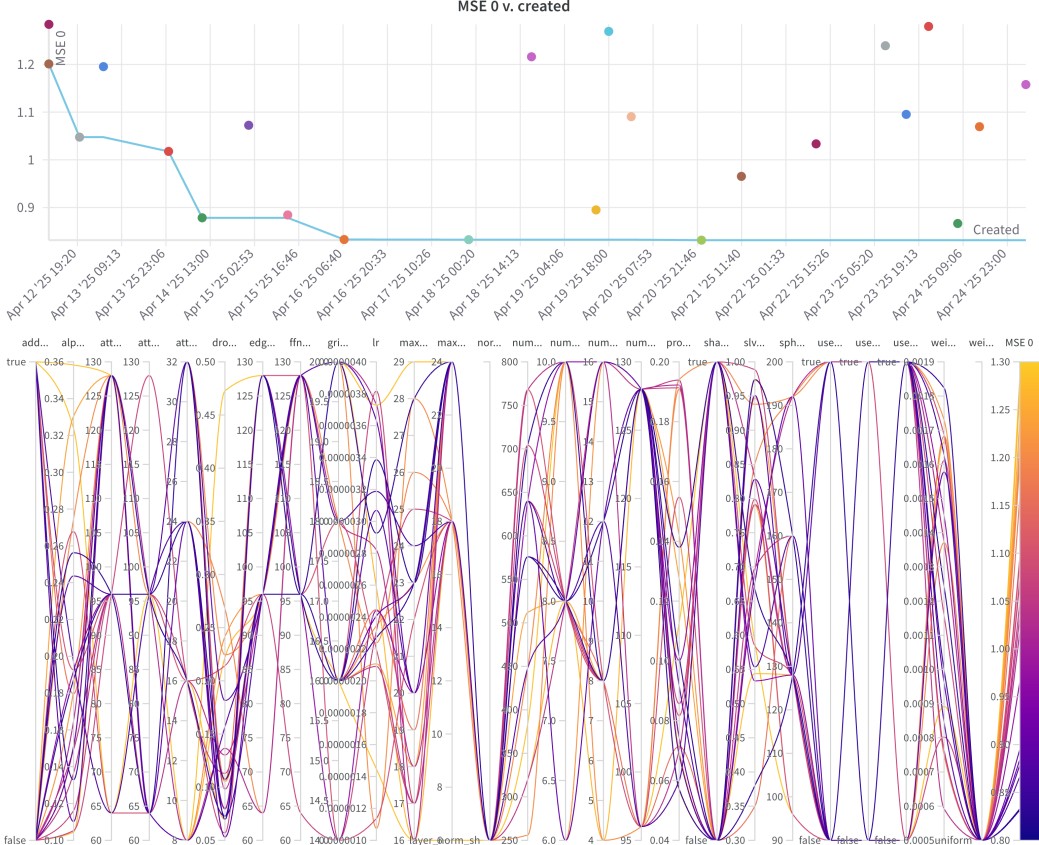

Figure 7: Hyperparameter tuning of Solvaformer. We ran a total of 23 different runs. WandB agents selected hyperparameters of successive runs using Bayesian optimization where performance was measured by MSE on the BigSolDB2.0 validation set.

link: https://zenodo.org/records/15094979
version: Published March 27, 2025 | Version v1

- CombiSolv-QM (Vermeire & Green, 2020):
link: https://zenodo.org/records/5970538
version: Published July 1, 2022 | Version v1.2

