# OpenReview forum: "Solvaformer: Minimizing Geometric Redundancy for Scalable Solubility Prediction"
_ICLR.cc/2026/Workshop/GRaM — ICLR 2026 Workshop GRaM Poster_

### Official Review · Reviewer_sPnc · 2026-02-10
**The paper is clear and presents an interesting idea about whether full geometric equivariance is needed for solubility prediction. It fits the workshop theme.**

**Rating:** 7
**Confidence:** 3

**Review:**

The paper presents a clear and well-motivated idea that fits the workshop theme of scale and simplicity. It raises an interesting question about whether full SE(3)-equivariance is necessary for solubility prediction and explores this empirically using two contrasting model designs. For a Tiny Paper, the contribution is appropriate, and the results are clearly presented.

However, the paper could better explain why equivariance may not be needed, beyond showing empirical performance, for example, it is unclear whether the authors believe that standard graph neural networks already capture sufficient geometric structure implicitly, or the relevant geometric information for solubility is largely encoded in invariant electronic descriptors, making explicit equivariant reasoning unnecessary.

Pros:

- Clear and focused research question aligned with the workshop theme.
- Simple and well-motivated model design.
- Empirical results support the main claim of possible unnecessity of equivariance information.

Cons:

- Limited intuition on why equivariance is unnecessary in this setting.
- The discussion of scalability is mostly qualitative, which is not supported by quantitative comparisons.

The paper could be strengthened by adding intuition on the role of equivariance and by including simple computational cost comparisons to support scalability claims.

**Pmlr Suitability:**

NA

---

### Official Review · Reviewer_QJZA · 2026-02-13
**A balanced, scalable approach to geometric redundancy in solubility prediction**

**Rating:** 6
**Confidence:** 5

**Review:**

This paper attempts to challenge the common assumption that more equivariance is always better, arguing that full SE(3)-equivariance in dynamic, multi-component solvent systems introduces unnecessary "geometric redundancy" and computational bloat. Solvaformer is a graph transformer that selectively grounds only local interactions in geometry while utilizing a standard Transformer backbone for global context. I found this approach to be technically mature and a welcome departure from computationally expensive global equivariant models. The results on solubility benchmarks show that the model captures the necessary physics with significantly less overhead. However, a major limitation is the narrowness of the evaluation; while solubility is a great test case, it remains unclear if this "selective grounding" would fail in more sensitive tasks like protein-ligand docking, where global orientation is more than just "redundant."  Despite this, the paper offers a high-impact, pragmatic strategy for scalable molecular modeling that is perfectly suited for a workshop discussion.

**Pmlr Suitability:**

NA

---

### Official Review · Reviewer_HEHv · 2026-02-16
**Study of geometric redundancy in solubility prediction**

**Rating:** 5
**Confidence:** 3

**Review:**

The paper presents Solvaformer, a hybrid graph transformer that applies SE(3)-equivariant attention to intramolecular interactions while using scalar attention for intermolecular solute–solvent interactions, aiming to reduce geometric redundancy in solubility prediction. The model is evaluated against several baselines, including an MPNN augmented with physics-informed partial charges, which slightly outperforms Solvaformer

Strengths:
- The focus on reducing geometric redundancy fits well with the workshop’s theme around scaling and simplicity
- Clear architectural motivation and ablation studies
- Good to see honest reporting on simpler baselines.
- Interpretable attention analysis with chemical insight

Weaknesses:
- Train and test sets have very similar target distributions, limiting evaluation under distribution shift
- Results appear to be reported from single runs only; I would have liked to see statistical validation across seeds
- No runtime or computational cost analysis, despite scalability claims

The paper offers a clear and well-motivated take on geometric redundancy in solubility prediction. That said, the experimental section feels somewhat limited: there’s no statistical validation, no evaluation under distribution shift, and no comparison of computational cost, which I think would be important to include. Based on the presented evidence, I’m not fully convinced by the scalability claims.

**Pmlr Suitability:**

NA

---

### Official Review · Reviewer_qTsD · 2026-02-22
**Hybrid Equivariance for Solute–Solvent Modeling: Clear and Promising, but Missing Key Validations**

**Rating:** 6
**Confidence:** 4

**Review:**

This paper proposes Solvaformer, a hybrid architecture for solute–solvent systems that uses SE(3)-equivariant attention for intramolecular interactions and computationally efficient scalar cross-attention for intermolecular interactions. The model is trained with an alternating-batch multi-task setup combining experimental solubility data (BigSolDB 2.0) and QM-derived solvation free energy data (CombiSolv-QM). The central motivation is that relative solute–solvent geometry in solution is largely disordered, so enforcing full intermolecular SE(3) equivariance may be unnecessary.

Strengths:
1) Clear writing and presentation: the motivation, method design choices, and experimental setup are cleanly communicated and are easy to follow.

2) Thorough empirical evaluation: includes meaningful ablations and comparisons that help interpret where performance gains come from.

3) Simple and intuitive design: the separation of intra- vs. inter-molecular modeling is conceptually clean and likely practical for implementation and scaling.

Weaknesses:

1) 3D geometry quality and robustness are under-explored: conformers are generated simply via RDKit/MMFF, which may deviate from solution-phase conformational ensembles. The paper would benefit from robustness/sensitivity analyses to assess the impact of geometric noise.

2) Key claim lacks direct empirical validation: while the paper argues that intermolecular SE(3) equivariance is unnecessary, it does not provide a convincing “accuracy vs. cost” comparison or a direct test of the assumption. This is instead deferred to future work.

3) Limited evidence for broader impact/generalization: the empirical evidence is mostly limited to solubility/solvation benchmarks, so the broader applicability remains somewhat under-supported.

**Pmlr Suitability:**

NA

---

### Meta-Review · Area_Chair_PQ8Q · 2026-02-26

**Decision:**

Accept

**Metareview:**

The reviewers agree that the fits the theme of the workshop and presents interesting contributions. The evaluation is however mentioned as a limitation that the authors are encouraged to address.

**Relevance To Proceedings:**

Tiny paper — does not apply

**Relevance To Workshop:**

Yes — suitable for GRaM

---

### Decision · Program_Chairs · 2026-03-02

Accept (Poster)